# Hybrid Neural Autoencoders for Stimulus Encoding in Visual and Other Sensory Neuroprostheses

**Jacob Granley**
Department of Computer Science
University of California, Santa Barbara
jgranley@ucsb.edu

**Lucas Relic**
Department of Computer Science
University of California, Santa Barbara
lucasrelic@ucsb.edu

**Michael Beyeler**
Department of Computer Science
Department of Psychological & Brain Sciences
University of California, Santa Barbara
mbeyeler@ucsb.edu

## Abstract

Sensory neuroprostheses are emerging as a promising technology to restore lost sensory function or augment human capabilities. However, sensations elicited by current devices often appear artificial and distorted. Although current models can predict the neural or perceptual response to an electrical stimulus, an optimal stimulation strategy solves the inverse problem: what is the required stimulus to produce a desired response? Here, we frame this as an end-to-end optimization problem, where a deep neural network stimulus encoder is trained to invert a known and fixed forward model that approximates the underlying biological system. As a proof of concept, we demonstrate the effectiveness of this hybrid neural autoencoder (HNA) in visual neuroprostheses. We find that HNA produces high-fidelity patient-specific stimuli representing handwritten digits and segmented images of everyday objects, and significantly outperforms conventional encoding strategies across all simulated patients. Overall this is an important step towards the long-standing challenge of restoring high-quality vision to people living with incurable blindness and may prove a promising solution for a variety of neuroprosthetic technologies.

## 1 Introduction

Sensory neuroprostheses are emerging as a promising technology to restore lost sensory function or augment human capacities [1, 2]. In such devices, diminished sensory modalities (e.g., hearing [3], vision [4, 5], cutaneous touch [6]) are re-enacted through streams of artificial input to the nervous system. For example, visual neuroprostheses electrically stimulate neurons in the early visual system to elicit neuronal responses that the brain interprets as visual percepts. Such devices have the potential to restore a rudimentary form of vision to millions of people living with incurable blindness.

However, all of these technologies face the challenge of interfacing with a highly nonlinear system of biological neurons whose role in perception is not fully understood. Due to the limited resolution of electrical stimulation, prostheses often create neural response patterns foreign to the brain. Consequently, sensations elicited by current sensory neuroprostheses often appear artificial and distorted [7, 8]. A major outstanding challenge is thus to identify a stimulus encoding that leads to perceptually intelligible sensations. Often there exists a forward model, $f$ (Fig. 1A), constrained by empirical data, that can predict a neuronal or (ideally) perceptual response to the applied stimulus (see [9] for a recent review). To find the stimulus that will elicit a desired response, one essentially needs to find

36th Conference on Neural Information Processing Systems (NeurIPS 2022).

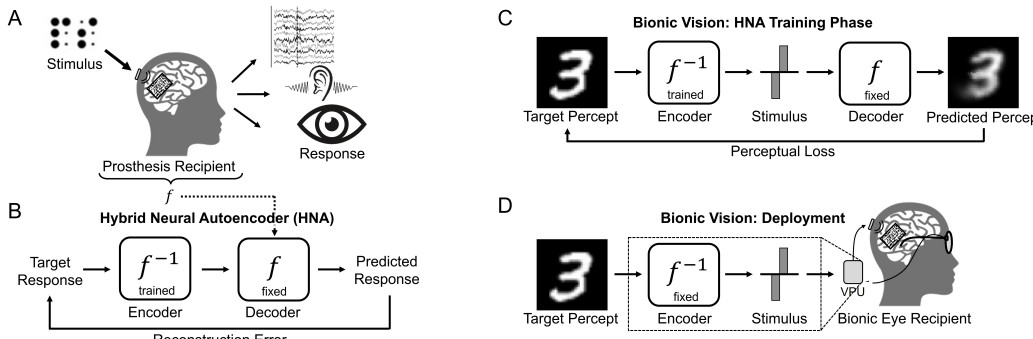

Figure 1: *A)* Sensory neuroprosthesis. A forward model ($f$) is used to approximate the neuronal or, ideally, perceptual response to electrical stimuli. *B)* Hybrid neural autoencoder (HNA). A deep neural encoder ($f^{-1}$) is trained to predict the patterns of electrical stimulation that elicit responses closest to the target. *C)* Visual neuroprostheses are one prominent application of HNA, where an encoder can be trained to predict the electrical stimulation needed to elicit a desired visual percept. *D)* The trained encoder is deployed on a vision processing unit (VPU), predicting stimuli in real-time that are decoded by the patient's visual cortex.

the inverse of the forward model, $f^{-1}$. However, realistic forward models are rarely analytically invertible, making this a challenging open problem for neuroprostheses.

Here we propose to approach this as an end-to-end optimization problem, where a deep neural network (DNN) (*encoder*) is trained to invert a known, fixed forward model (*decoder*, Fig. 1B). The encoder is trained to predict the patterns of electrical stimulation patterns that elicit perception (*e.g.,* vision, audition) or neural responses (*e.g.,* firing rates) closest to the target. This hybrid neural autoencoder (HNA) could in theory be used to optimize stimuli for any open-loop neuroprosthesis with a known forward model that approximates the underlying biological system.

In order to optimize end-to-end, the forward model must be differentiable and computationally efficient. When this is not the case, an alternative approach is to train a surrogate neural network, $\hat{f}$, to approximate the forward model [10–13]. However, even well-trained surrogates may result in errors when included in our end-to-end framework, due to the encoders' ability to learn to exploit the surrogate model. We therefore also evaluate whether a surrogate approach to HNA is suitable.

To this end, we make the following contributions:

- We propose a hybrid neural autoencoder (HNA) consisting of a deep neural encoder trained to invert a fixed, numerical or symbolic forward model (decoder) to optimize stimulus selection. Our framework is general and addresses an important challenge with sensory neuroprostheses.

- As a proof of concept, we demonstrate the utility of HNA for visual neuroprostheses, where we predict electrode activation patterns required to produce a desired visual percept. We show that the HNA is able to produce high-fidelity, patient-specific stimuli representing handwritten digits and segmented images of everyday objects, drastically outperforming conventional approaches.

- We evaluate replacing a computationally expensive or nondifferentiable forward model with a surrogate, highlighting benefits and potential dangers of popular surrogate techniques.

## 2  Background

**Sensory Neuroprostheses**   Recent advances in neural understanding, wearable electronics, and biocompatible materials have accelerated the development of sensory neuroprostheses to restore perceptual function to people with impaired sensation. Use of neuroprostheses typically requires invasive implants that elicit neural responses via electrical, magnetic, or optogenetic stimulation. Two of the most promising applications are cochlear implants, which stimulate the auditory nerve to elicit sounds [3], and visual implants (see next subsection) to restore vision to the blind. However, a variety of other devices are in development; for instance, to restore touch [6, 14] or motor function [15]. The latter differ from other sensory neuroprostheses in that they generate stimuli using motor

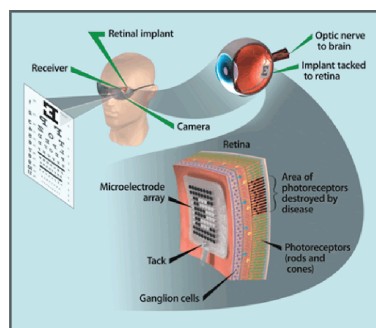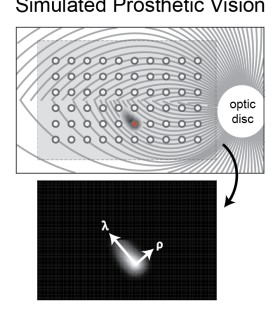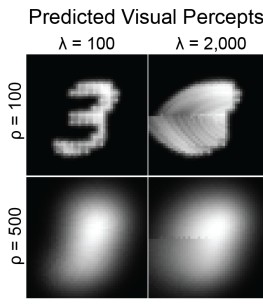

Figure 2: *Left*: Visual prosthesis. Incoming target images are transmitted from a camera to an implant in the retina, which encodes the image as an electrical stimulus pattern. *Center*: Electrical stimulation (red disc) of a nerve fiber bundle (gray lines) leads to elongated tissue activation (gray shaded region) and phosphenes whose shape can be described by two parameters, $\lambda$ (axonal spread) and $\rho$ (radial spread). *Right*: Predicted percepts for an MNIST digit using varying $\rho$ and $\lambda$ values.

feedback (*closed loop*) [16, 17]. In the absence of feedback (*open loop*), a proper stimulus encoding is paramount to the success of these devices.

**Restoring Vision to the Blind**    For millions of people who are living with incurable blindness, a visual prostheses (*bionic eye*, Fig. 2, *left*) may be the only treatment option [18]. Analogous to cochlear implants, these devices electrically stimulate surviving cells in the visual pathway to evoke visual percepts (*phosphenes*), which can support simple behavioral tasks [5, 19, 20].

A common misconception is that each electrode in the array can be thought of as a pixel in an image; to generate a complex visual experience, one then simply needs to turn on the right combination of pixels [21]. However, recent evidence suggests that phosphenes often appear distorted (*e.g.*, as lines, wedges, and blobs) and vary drastically across subjects and electrodes [4, 7].

Phosphene appearance has been best characterized in epiretinal implants, where inadvertent activation of nerve fiber bundles (NFBs) in the optic fiber layer of the retina leads to elongated phosphenes [22, 23] (Fig. 2, *center*). To this end, Granley *et. al* [24] developed a forward model to predict phosphene shape as a function of both neuroanatomical parameters (*i.e.*, location of the stimulating electrode) and stimulus parameters (*i.e.*, pulse frequency, amplitude, and duration). With this model, phosphenes are primarily characterized by two main parameters, $\rho$ and $\lambda$, which dictate the size and elongation of the elicited phosphene, respectively (Fig. 2, *right*). These parameters can be determined using psychophysical tasks (*e.g.*, drawings, brightness ratings) [22, 24], and although they vary drastically across patients [22], they do not change much over time [25, 26]. Stimulation from multiple electrodes is nonlinearly integrated into a combined perception, and if two electrodes happen to activate the same NFB, they might not generate two distinct phosphenes.

## 3   Related Work

The conventional 'naive' encoding strategy sets the amplitude of each electrode to the brightness of the corresponding pixel in the target image [5, 27], making the stimulus a down-sampled version of the target. Although simple, this strategy only works with near-linear forward models, cannot account for real phosphene data, and often leads to unrecognizable percepts (Fig. 2, *right*) [7, 22].

Many alternative stimulation strategies have been proposed [28]. Shah *et al.* [29] used a greedy approach to iteratively select the stimuli that best recreate a desired neural activity pattern over a given temporal window, assuming that the brain would integrate them into a coherent visual percept. Ghaffari *et al.* [30] used a neural network surrogate model combined with an interior point algorithm to optimize for localized, circular neural activation patterns for individual electrodes. Fauvel *et al.* [31] used human in-the-loop Bayesian optimization to achieve encodings perceptually favored by the simulated patient. Spencer *et al.* [32] proposed framing stimulus encoding as inversion of a forward model of neural activation patterns, but to approximate the inverse, their approach either requires simplification or is NP-hard [32].

Van Steveninck *et al.* [33] proposed an end-to-end optimization strategy with a fixed phosphene model, similar to HNA. However, their approach crucially differs from ours in its inclusion of a secondary DNN to post-process the predicted phosphenes. This is a critical limitation, because a low reconstruction loss does not reveal whether a high-fidelity encoder was learned or whether the secondary decoder network simply learned to correct for the encoder's mistakes. In addition, they used an unrealistic phosphene model that simply upscales and smooths a binary stimulus pattern. It is therefore not clear whether their results would generalize to real visual prosthesis patients.

Relic *et al.* [10] also utilized the end-to-end approach, but without the secondary decoder network used in [33]. They used a more realistic phosphene model, which accounts for some spatial distortions (*e.g.*, axonal streaks), but not the effects of stimulus parameters. Since including a realistic phosphene model in the loop is not straightforward, they instead trained a surrogate neural network to approximate the forward model. We re-implemented Relic's surrogate approach in this paper as a baseline method to compare against, as described in Section 4.

Taken together, we identified three main limitations of previous work that this study aims to address:

1) **Unrealistic forward models.** Most previous approaches (*e.g.*, [29, 32, 33]) use an overly simplified forward model that cannot account for empirical data [7, 22]. We overcome this limitation by developing (and inverting) a differentiable version of a neurophysiologically informed and psychophysically validated phosphene model [24] that can account for the effects of stimulus amplitude, frequency, and pulse duration on phosphene appearance.

2) **Optimization of neural responses.** Most previous approaches (*e.g.*, [29, 32]) focus on optimizing neural activation patterns in the retina in response to electrical stimulation ("bottom-up"). However, the visual system undergoes extensive remodeling during blinding diseases such as retinitis pigmentosa [34]. Thus the link between neural activity and visual perception is unclear. We overcome this limitation by inverting a phenomenological ("top-down") model constrained by behavioral data that predicts visual perception directly from electrical stimuli [22, 24].

3) **Objective function** Most previous approaches rely on minimizing mean squared error (MSE) between the target and reconstructed image. Although simple and efficient, MSE is known to be a poor measure of perceptual dissimilarity for images [35] and does not align well with human assessments of image quality [36]. We overcome this limitation by proposing a joint perceptual metric that combines mean absolute error (MAE), VGG, and Laplacian smoothing losses.

## 4 Methods

**Problem Formulation**   We consider a system where there is some known forward process $f$ mapping stimuli to responses $f : \mathcal{S} \mapsto \mathcal{R}, f(\mathcal{S}) \subset \mathcal{R}$. In the case of visual prostheses, $f$ may map stimuli to visual percepts. $f$ may optionally be parameterized by patient-specific parameters $\phi$.

Finding the best stimulus for an arbitrary target response $\mathbf{t} \in \mathcal{R}$ is equivalent to finding the inverse of $f$. However, since not every response can be perfectly reproduced by a stimulus, the true inverse of $f$ is not well defined. We therefore seek the pseudoinverse (still denoted as $f^{-1}$ for simplicity) instead, which outputs the stimuli that produce the closest response to $\mathbf{t}$ under some distance metric $d$:

$$f^{-1}(\mathbf{t}, \phi) := \arg\min_{\mathbf{s} \in \mathcal{S}} d(f(\mathbf{s}; \phi), \mathbf{t}). \tag{1}$$

This problem is straightforward to solve using an autoencoder approach, where a learned encoder $f^{-1}$ is trained to invert the fixed decoder $f$ (i.e., forward model).

**Encoder**   We approximate the pseudoinverse $f^{-1}$ with a DNN encoder $\hat{f}^{-1}(\mathbf{t}, \phi; \theta)$ with weights $\theta$, trained to minimize the distance $d$ between the target image $\mathbf{t}$ and predicted image $\hat{\mathbf{t}}$:

$$\min_{\theta, \, \phi \sim p(\phi)} d(\mathbf{t}, \hat{\mathbf{t}}) \tag{2}$$

$$\hat{\mathbf{t}} = f(\hat{f}^{-1}(\mathbf{t}, \phi; \theta); \ \phi), \tag{3}$$

where $\phi$ is sampled from a uniform random distribution spanning the empirically observed range of patient-specific parameters [22, 24].

We use a simple architecture consisting solely of fully connected (FC) and batch normalization (BN) [37] layers (1.4M trainable parameters). First, the target $\mathbf{t}$ is flattened and input to a FC layer. In parallel, the patient parameters $\phi$ are input to a BN layer and two hidden FC layers. Then, the outputs of these two paths are concatenated, and the combined vector fed through two FC layers, producing an intermediate representation $\mathbf{x}$. Amplitudes are predicted from $\mathbf{x}$ with a FC layer. The amplitudes are then concatenated with $\mathbf{x}$, put through a BN layer, and used to predict frequency and pulse duration, each with a FC layer. The outputs are merged into a stimulus matrix $\hat{\mathbf{s}}$. All layers use leaky ReLU activation, except for stimulus outputs, which use ReLU to enforce nonnegativity.

**Decoder**  The HNA decoder is a differentiable approximation of the underlying biological system, and describes the transform from stimulus to response. For our decoder $f$, we use a reformulated but equivalent version of the model described in [24]. This model is specific to epiretinal prostheses; analogous models exist for other neuroprostheses (*e.g.*, auditory [38–43], tactile and somatosensory [44–48]), and could potentially be adapted for use with HNA. We use a square $15 \times 15$ array of $150\mu m$ electrodes, spaced $400\mu m$ apart and centered on the fovea. The size and scale of this device are motivated by similar designs in real epiretinal implants.

$f$ takes as input a stimulus matrix $\mathbf{s} \in \mathbb{R}^{n_e \times 3}_{\geq 0}$, where the stimulus on each electrode ($\mathbf{s_e}$) is a biphasic pulse train described by its frequency, amplitude, and pulse duration. A simulated map of retinal NFBs is used to predict phosphene shape. Following [22], each retinal ganglion cells' activation is assumed to be the maximum stimulation intensity along its axon. Axon sensitivity is assumed to decay exponentially with i) distance to the stimulating electrode (radial decay rate, $\rho$) and distance to the soma along the curved axon (axonal decay rate, $\lambda$). To account for stimulus properties [24], $\rho$, $\lambda$, and the per-electrode brightness are scaled by three simple equations $F_{\text{size}}(\mathbf{s_e}, \phi)$, $F_{\text{streak}}(\mathbf{s_e}, \phi)$, and $F_{\text{bright}}(\mathbf{s_e}, \phi)$, respectively.

The brightness of a pixel located at the point $\mathbf{x} \in \mathbb{R}^2$ in the output image is given by

$$f(\mathbf{s}; \phi) = \max_{\mathbf{a} \in A} \sum_{e \in E} F_{\text{bright}}(\mathbf{s_e}, \phi) \exp \left( \frac{-||\mathbf{x} - \mathbf{e}||_2^2}{2\rho^2 F_{\text{size}}(\mathbf{s_e}, \phi)} + \frac{-d_s(\mathbf{x}, \mathbf{a})^2}{2\lambda^2 F_{\text{streak}}(\mathbf{s_e}, \phi)} \right) \quad (4)$$

where $A$ is the cells' axon trajectory, $E$ is the set of electrodes, $\phi = \{\rho, \lambda, ...\}$ is a set of 12 patient-specific parameters, and $d_s$ is the path length along the axon trajectory [49]from $\mathbf{a}$ to $\mathbf{x}$:

$$d_s(\mathbf{x}, \mathbf{a}) = \int_{\mathbf{a}}^{\mathbf{x}} \sqrt{A(\theta)^2 + \left( \frac{dA(\theta)}{d\theta} \right)^2} \, d\theta. \quad (5)$$

This model ($f$) can be fit to individual patients; however, it is computationally slow and not differentiable. For more details on these equations, see [24]. We therefore considered two approaches:

- **Differentiable Model:** We reformulated equations 4 and 5 into an equivalent set of parallelized matrix operations, implemented in Tensorflow [50]. Significant efforts were put towards developing a model optimized for XLA engines on GPU, resulting in speedups of up to 5000x compared to the model as presented in [24], enabling large-scale gradient descent. To enforce differentiability, we numerically approximated $d_s$ using $|A| = 500$ axon segments per axon.
- **Surrogate Model:** We also implemented the surrogate approach from [10] as a baseline method, where $f$ is approximated with another DNN $\hat{f}_\phi(\mathbf{s}; \theta_f)$ with weights $\theta_f$. To achieve this we generated 50,000 percepts using randomly selected stimuli passed through $f$ and fit a DNN to produce identical images. As $f$ is highly dependent on patient-specific parameters $\phi$, we generated new data and fit a separate surrogate model for each $\phi$ in our experimental set. Specific implementation details of the surrogate are presented in Appendix A. Our implementation improves upon [10] by using the more advanced phosphene model described above, which accounts for effects of stimulus properties and allows for optimization of stimulus frequency in addition to amplitude.

**Metrics**  To measure perceptual similarity, we use a joint perceptual objective consisting of a VGG [51] similarity term, a mean absolute error (MAE) term, and a smoothness regularization term. The MAE term is given by $L_{\text{MAE}} = \frac{1}{|\mathbf{t}|}||\mathbf{t} - \hat{\mathbf{t}}||_1$.

The VGG term aims to capture higher-level differences between images [33, 52]. The target image and reconstructed phosphene are input to VGG-19 pretrained on ImageNet [53], and the MSE between

the activations on a downstream convolutional layer is computed. Let $V_l$ be a function that extracts the activations of the $l$-th convolutional layer for a given image. The VGG loss is then defined as $L_{\text{VGG}} = \frac{1}{|\mathbf{t}|}||V_l(\mathbf{t}) - V_l(\hat{\mathbf{t}})||_2^2$.

We also include a smoothing regularization term. This term imposes a loss on the second spatial derivative of the predicted image. A low second derivative implies that where the target image does change, it changes slowly. We found this encouraged smoother, more connected phosphenes. To approximate the second derivative, we convolve the image with a Laplacian filter [54] of size $k$, denoted by $Lap(\cdot, k)$, and compute the mean. The smoothness loss is given by:

$$L_{\text{Smooth}} = \frac{1}{|\hat{\mathbf{t}}|} \sum_i \left( \frac{\partial^2}{dx^2} \hat{\mathbf{t}} \right)_i = \frac{1}{|\hat{\mathbf{t}}|} \sum_i Lap(\hat{\mathbf{t}}, k)_i. \tag{6}$$

Our final objective is the weighted sum of the three individual losses, given by Eq. 7, where $\alpha$ and $\beta$ are hyperparameters controlling the relative weighting of the three terms.

$$d = L_{\text{MAE}} + \alpha L_{\text{Smooth}} + \beta L_{\text{VGG}}. \tag{7}$$

We also implement a secondary metric to quantify phosphene recognizability, applicable only for the MNIST reconstruction task. We first pre-train a classifier network on the MNIST targets until it reaches 99% test accuracy, and then fix the weights. The relative accuracy (RA) is then defined as the ratio of the classifiers accuracy on the reconstructed images to its accuracy on the targets $RA = ACC/ACC(\mathbf{t})$. A perfect encoder would result in $RA = 100\%$. A similar process was not possible for the COCO task due to the possibility of having multiple objects in each target image.

**Training/Optimization**   We trained using Tensorflow 2.7 [50] on a single NVIDIA RTX 3090 with 24GB memory. Stochastic gradient descent with Nesterov momentum was used to minimize the joint perceptual loss. We used a batch size of 16 due to memory limitations imposed by $f$. The amplitude, frequency predictions are scaled by 2, 20 respectively, while the pulse duration predictions were shifted by 1e-3 prior to being fed through the decoder. This encourages the initial predictions of the network to be in a reasonable range. The Laplacian filter size $k$ is set to 5. We choose $l$ to be first convolutional layer in the last block using cross validation (see Appendix B). Similarly, we perform cross validation to find the best values for $\alpha$ and $\beta$. Instead of using one value, we found that incrementally increasing the weighting of the VGG loss ($\beta$) from 0 while simultaneously decreasing the initially high weight on the smoothing constraint ($\alpha$) was crucial for performance, especially when the range of allowed $\phi$ values was large (see Appendix B).

**Datasets**   We first evaluated on handwritten digits from MNIST [55], enabling comparison to previous works [10]. Images preprocessing consisted of resizing the target images to the same shape as the output of $f$ (49x49). We also evaluate on more realistic images of common objects from the MS-COCO [56] dataset. We selected a subset of 25 of the MS-COCO object categories deemed more likely to be encountered by blind individuals (*e.g.* people, household objects), and use only images that contain at least one instance of these objects. We further filter out images by various other criteria, such as being too cluttered or too dim. This process results in a total of approximately 47K training images and 12K test images. See Appendix C for a full description of the selection process.

Natural images often contain too much detail to be encoded with prosthetic vision. While scene simplification strategies exist [57], we focus on the encoding algorithm, so we simply used the ground-truth segmentation masks to segment out the objects of interest. The images were then converted to grayscale, and resized to $49 \times 49$ pixels.

## 5   Results

### 5.1   MNIST

The phosphenes produced from the HNA, surrogate, and naive encoders on the MNIST test set are shown in Fig. 3 and performance is summarized in Table 1. For each MNIST sample, the target image is input to the encoder, which predicts a stimulus. The stimulus is fed through the true forward model $f$, and the predicted phosphene is shown. Since the surrogate method must be retrained for each $\phi$, results are only shown for 4 simulated patients. Our proposed approach outperformed the baselines across all metrics (see Appendix D for a comparison of stimuli).

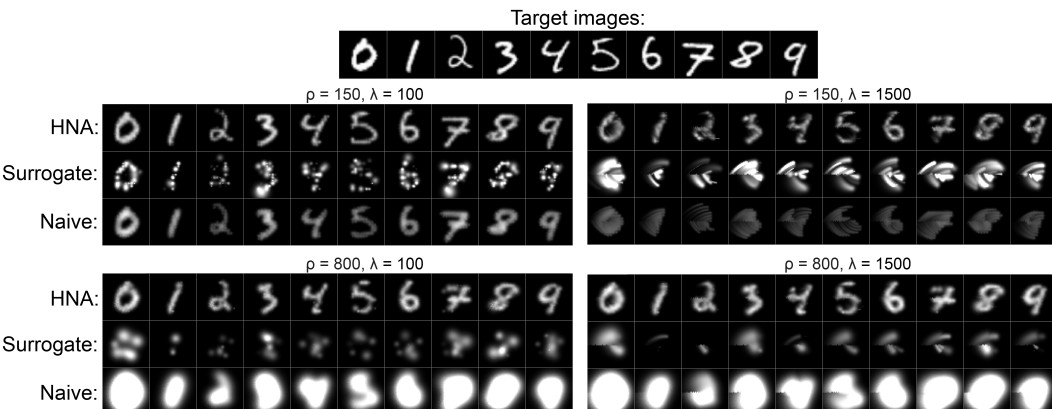

Figure 3: Reconstructed MNIST targets for HNA, surrogate, and naive encoders across 4 specific simulated patients. Note that the brightness of the naive encoder is clipped for display

Table 1: MNIST performance

| Encoding | $\rho$=150 $\lambda$=100 | | | $\rho$=150 $\lambda$=1500 | | | $\rho$=800 $\lambda$=100 | | | $\rho$=800 $\lambda$=1500 | | |
|---|---|---|---|---|---|---|---|---|---|---|---|---|
| | Joint Loss | MAE | RA | Joint Loss | MAE | RA | Joint Loss | MAE | RA | Joint Loss | MAE | RA |
| Naive | 1.161 | 0.1855 | 90.3 | 1.442 | 0.214 | 78.1 | 8.152 | 1.500 | 34.8 | 8.780 | 1.726 | 28.8 |
| Surrogate | 2.509 | 0.1351 | 53.8 | 3.118 | 0.2431 | 30.7 | 1.692 | 0.2135 | 19.9 | 1.694 | 0.2237 | 18.1 |
| HNA | **0.559** | **0.064** | **98.1** | **1.029** | **0.1412** | **89.3** | **0.913** | **0.113** | **95.9** | **0.957** | **0.126** | **94.8** |

## 5.2 COCO

The phosphenes produced by HNA and the naive encoder for the segmented COCO dataset are shown in Fig. 4. We omit the surrogate results due to its poor perceptual performance on MNIST. Averaged across all $\phi$, HNA had a joint loss of 0.713 on the test set and MAE of 0.1408, while the naive encoder had a joint loss of 1.873 and MAE of 0.2830.

## 5.3 Modeling Patient-to-Patient Variations

MNIST encoder performance across simulated patients ($\phi$) is shown in Fig. 5. Since the surrogate encoder has to be retrained for each patient, comparison is infeasible. To visualize the effects of changing $\rho$ and $\lambda$ on the produced phosphenes, Fig. 5A shows the result of encoding two example MNIST digits, both using the naive method and our encoder. As $\lambda$ increases, the naive phosphenes appear increasingly elongated, and as $\rho$ increases, the phosphenes become increasingly large and blurry. The phosphenes from HNA are slightly too dim and disconnected at low $\rho$, but are relatively stable across other values of $\rho$ and $\lambda$.

To compare performance across the entire dataset, we computed the average test set loss across the same range of $\rho$ and $\lambda$ (Fig. 5B). The encoder performs well across a wide range of simulated patients, with larger loss only at low $\rho$. The naive method performs well only on a limited set of $\phi$, with small $\lambda$ and $\rho \approx 200$. The naive loss was higher than the learned encoder at every simulated point. Random sampling of $\rho$ and $\lambda$ for each image results in a joint loss of 0.921, MAE of 0.120, and RA of 94.0% for HNA, while the naive encoder results in a joint loss of 3.17, MAE of 0.596, and RA of 63.6%. The same analysis yielded similar results on COCO (Appendix E). An analysis across other parameters is presented in Appendix F.

In order for prosthetic vision to be useful, different instances of the same objects would ideally produce similar phosphenes, allowing for consistent perception. To evaluate whether our model achieves this, we cluster the target images and resulting phosphenes using t-SNE [58] shown in Fig. 5C. The ground truth images form clusters corresponding to the digits 0-9. The phosphenes from our encoder roughly form similar, slightly less separated groupings, whereas the naive phosphenes do not. To ensure that this was not the result of bad t-SNE hyperparameters, we repeated the clustering across different perplexities and learning rates, obtaining similar or worse results.

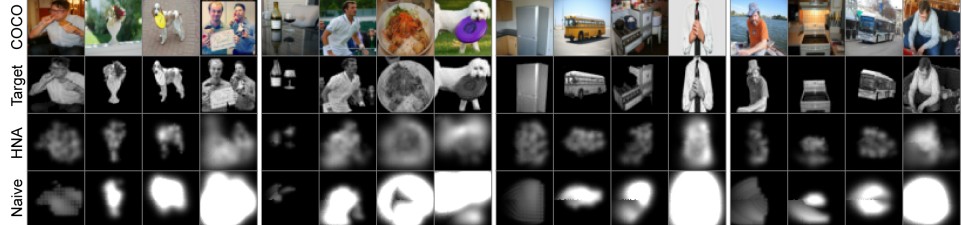

Figure 4: Original (*top row*), segmented (*second row*), and reconstructed targets for the COCO dataset, for both HNA (*third row*) and naive encoders (*bottom row*). Left to right within each block of 4 images, $\rho$ takes values of 200, 400, 600, 800. Left to right across blocks, $\lambda$ takes values of 250, 750, 1250, 2000. Note that the brightness of the naive method is clipped for display.

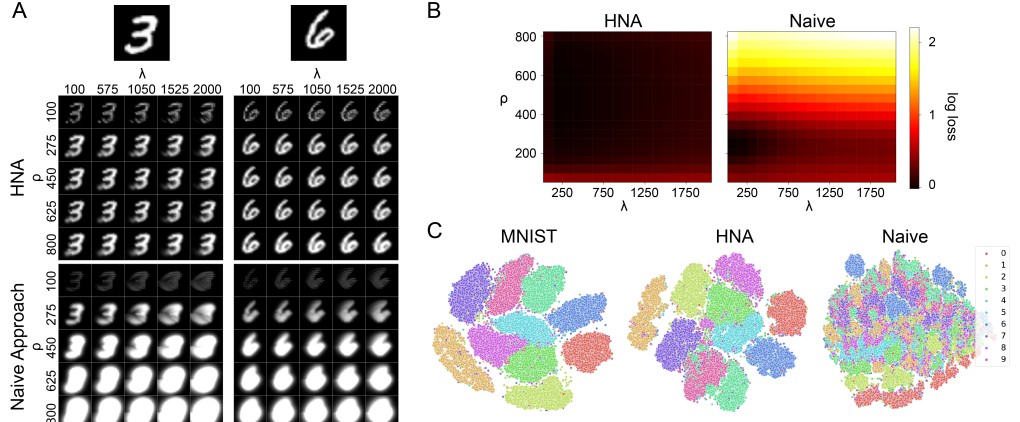

Figure 5: Encoder performance across simulated patients (varying $\rho$ and $\lambda$) on the MNIST dataset. *A*: Target, HNA encoder, and naive encoder phosphenes for two example digits. *B*: Heatmaps showing the log joint loss across $\rho$ and $\lambda$ for HNA and naive encoders. *C*: T-SNE clusterings on original MNIST targets, HNA reconstructed phosphenes, and naive reconstructed phosphenes.

### 5.4 Joint Perceptual Error Ablation Study

To show that the joint perceptual metric performs better than any of its individual components, we train models using just the VGG loss and just MAE loss. Shown are values for $\rho$=150 and $\lambda$=600. As mentioned previously, encoders trained using just VGG loss fail to converge, thus we pretrain the VGG encoder using MAE and smoothing loss, then transition to using only VGG. We do not consider ablating the smoothing term (Eq. 6) because it is simply a regularization term. Fig. 6 shows the phosphenes produced by HNA trained on the joint, VGG-only, and MAE-only loss.

The VGG encoder had a test VGG loss of 4% lower than the joint model, but its produced phosphenes are oversmoothed and blurry. The MAE encoder had a final test MAE of 9% lower than the joint model, but its produced phosphenes are disconnected and low-quality. The joint model had a RA of 99.0%, the VGG encoder had a RA of 95.9%, and the joint model had a RA of 77.6%

## 6 Discussion

**Visual Prostheses** We found that HNA is able to produce high-fidelity stimuli from the MNIST and COCO datasets that outperform conventional encoding strategies across all tested conditions. Importantly, HNA produces phosphenes that are consistent across representations of the same object (Fig. 5C), which is critical to allowing prosthesis users to learn to associate certain visual patterns with specific objects. On the MNIST task, HNA produced high quality reconstructions, nearly matching the targets (Figure 3). On the harder COCO task, HNA significantly outperformed the naive encoder, but was still unable to capture all of the detail in the images. In Appendix G, we demonstrate that this is largely due to the implant's limited spatial resolution and not a fundamental limitation of HNA.

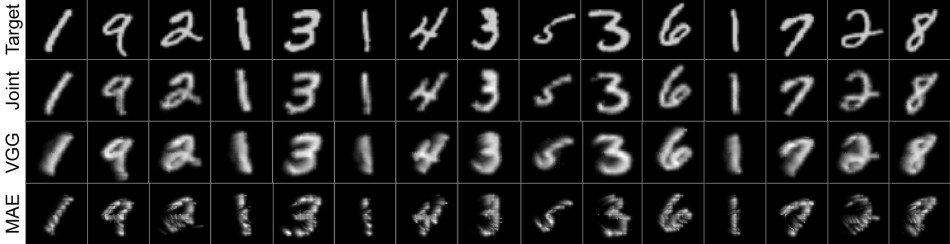

Figure 6: MNIST images for HNA encoders trained using the joint, VGG-only, and MAE-only loss.

Another advantage of the HNA is that it can be trained to predict stimuli across a wide range of patient-specific parameter values $\phi$, whereas the conventional naive encoder works well only for small values of $\rho$ and $\lambda$. This may be one reason why the naive encoding strategy has been shown to lead to substantial individual differences in visual outcomes [19, 59]. Our results suggest that stimuli produced with HNA may be able to reduce at least some amount of this patient-to-patient variability.

Furthermore, HNA also proved superior to a surrogate forward model. The latter offer an alternative when the forward model is computationally expensive or not differentiable. Understandably, any inaccuracies in the surrogate model will propagate to the learned encoder during training. However, we observed that even for well trained surrogates, the encoder may still learn to exploit the inexact surrogate instead of learning to invert the true model (see Appendix A). It is possible that this exploitation could be mitigated to some extent by adversarially-robust training techniques [60]. We suspect that the surrogate method's inferior performance here compared to [10] can be explained by our larger stimulus search space. Thus, we cannot currently suggest HNA for surrogate forward models, unless the forward model is sufficiently simple or has a small stimulus space.

**Deployment**   HNA encoders must be lightweight enough to be deployed in resource-limited neuroprosthetic environments. Our encoder's single image inference time was 1.2ms on GPU and 4ms on CPU. Future work could reduce these numbers through network pruning, mixed precision, and architecture search. Low-power Edge AI accelerators (*e.g.*, Intel's Neural Compute Stick) and dedicated neuromorphic hardware (*e.g.*, BrainChip's Akida SoC) may provide another solution.

**Broader Impacts**   While our work is presented in the context of visual prostheses, the HNA framework may apply to any sensory neuroprosthesis where stimulus selection can be informed by a numeric or symbolic forward model. For example, HNA could be used in cochlear implants [3] to choose stimuli that result in a desired sound, and in spinal cord implants [15] to find the best way to relay neural signals through a damaged section of the spinal cord. Conveniently, the forward models required by HNA have already been developed for a range of applications [38–48]. However, HNA might not apply to all neural interfaces, such as systems without a clear neural or perceptual target (*e.g.*, deep brain stimulation for the treatment of Parkinson's [61]) or closed-loop systems [16, 62].

**Limitations**   Despite HNA's potential, the current implementation has a number of limitations. First, as presented the HNA encoder only applies to static targets. Hence dynamic targets must be split into individual frames and encoded separately. However, one approach might be to encode entire stimulus sequences (instead of frames) that are optimized to reconstruct the dynamic target sequence.

Second, HNA works best if there is an accurate forward model mapping from stimulus space to perception. However, Appendix H shows that HNA may still give benefits over a naive encoding even when patient-specific parameters are unknown or mis-specified. In general, if a prosthesis elicits similar results across patients, then a non-patient-specific model would suffice.

Third, the current works deals only with simulated patients. The use of a DNN for stimulus encoding in real patients may raise safety concerns. Since we cannot examine the process by which stimuli are chosen, it is possible that HNA might produce harmful stimuli that could lead to serious adverse events (*e.g.*, seizures). However, this concern is mitigated by the fact that most neuroprostheses are equipped with firmware responsible for ensuring stimuli stay within FDA-approved safety limits.

## 7  Conclusion

In summary, this paper proposes a hybrid autoencoder structure as a general framework for stimulus optimization in sensory neuroprostheses and, as a proof of concept, demonstrates its utility on the prominent example of visual neuroprostheses, drastically outperforming conventional encoding strategies. This constitutes an important step towards the long-standing challenge of restoring high-quality vision to people living with incurable blindness and may prove a promising solution for a variety of neuroprosthetic technologies.

## 8  Acknowledgements

This work was supported by the National Institutes of Health (NIH R00 EY-029329 to MB).

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
