# OpenReview forum: "Hybrid Neural Autoencoders for Stimulus Encoding in Visual and Other Sensory Neuroprostheses"
_NeurIPS.cc/2022/Conference — NeurIPS 2022 Accept_

### Official Review · Reviewer_kHVG · 2022-07-11

**Rating:** 7
**Confidence:** 4
**Soundness:** 4 excellent
**Presentation:** 3 good
**Contribution:** 3 good

**Summary:**

The authors propose to use an autoencoder trained to invert a known feedforward model that approximates the biological network underlying the early human visual system.  They demonstrate the efficacy in application to visual prosthetics and show that there method leads electrode activation patterns that produce better and more intelligible decodings downstream in the biological system than comparable methods.  They claim their system to be generalizable to any type of sensory neuroprostheses (with obvious remodeling and retraining required).

**Questions:**


The author's described the limitations of the surrogate model and how even surrogates that perform well at the task have weakness that are able to be exploited by the encoding-decoding paradigm.
Do the author's think the failure of their surrogate model is possibly due to the low number of percept images used to constrain the deep neural network (50000 images may be enough to constrain for MNIST classification but as a downstream model of biological encoding does not seem to be nearly enough to tightly constrain filter the model)?

I know this is not the contribution of this work, but a bit more discussion of where the patient-specific data comes from in the text would be helpful for the reader.

**Limitations:**

The authors are upfront about the potential limitations and societal impacts of the work.

**Strengths And Weaknesses:**

They use a nice approach of treating the pseudo inverse of a known biological function as the encoder of an encoder decoder pair and learn the encoder given patient specific parameters.

The model of the biology, and treatment of the problem of phosphenes is very elegant, and works well.  It does suggest that generalizability of the methodology requires equally detailed biological forward models in other neuroprosthetic applications.

The results presented (especially in figure 3) are very impressive, but there should be more clear caveats that this methodology requires a patient specific model of phosphenes and different patients have quite different observed phosphenes.  Even with the reduction of patient to patient variability on MNIST introduced by this method, I would not anticipate as much success reducing variability on the already harder task of COCO like images.  It should also be noted that all of the evaluation here is done on a model of individual patients, and at no time are patients actually asked to observe these stimulation patterns and decode them downstream.

---

> ### Author Response · Authors · 2022-08-02
> **Author Response to Reviewer kHVG**
>
> We are grateful to the reviewer for their encouraging comments and insightful analysis.
>
> > *The model of the biology, and treatment of the problem of phosphenes is very elegant, and works well. It does suggest that generalizability of the methodology requires equally detailed biological forward models in other neuroprosthetic applications.*
>
> We agree that our method works best if there is an accurate forward model mapping from stimulus space to perception. However, HNA may still give some benefits over a naive encoding even when only a crude forward model is known. For example, Appendix H shows that HNA is still effective for visual prostheses even if patient-specific parameters are unknown or incorrect. In general, the benefits of increased accuracy from a more detailed model should be balanced with the added optimization complexity and increased compute time. We have updated the Discussion section to address these points (lines 304-309 and 316-319).
>
> > *... there should be more clear caveats that this methodology requires a patient specific model of phosphenes and different patients have quite different observed phosphenes.*
>
> The reviewer raises an excellent point. Although theoretically a patient-specific model is not a necessary HNA requirement, it is true that most current neuromodulation technologies lead to a wide variety of perceptual/behavioral outcomes. Hence a patient-specific model is expected to give the best HNA results. However, if a different prosthesis elicits similar results across patients, then a non-patient specific model would suffice. We consider it a strength of our approach that HNA can produce patient-specific encodings even in the more difficult case where the forward model varies drastically across patients. We have further expanded the Limitations section to address this issue (lines 316-319), and have also added Appendix H, which presents an analysis of the effects of incorrect or unknown patient parameters on the produced encoding.
>
> > *It should also be noted that all of the evaluation here is done on a model of individual patients, and at no time are patients actually asked to observe these stimulation patterns and decode them downstream.*
>
> We have added a sentence to the Limitations section of the Discussion emphasizing that these results are only for simulated patients (line 320).
>
> **Questions**
>
> > *The authors described the limitations of the surrogate model and how even surrogates that perform well at the task have weakness that are able to be exploited by the encoding-decoding paradigm. Do the author's think the failure of their surrogate model is possibly due to the low number of percept images used to constrain the deep neural network (50000 images may be enough to constrain for MNIST classification but as a downstream model of biological encoding does not seem to be nearly enough to tightly constrain filter the model)?*
>
> This is a sensible concern. It is important to note that each surrogate model is trained w.r.t. one set of patient-specific parameters, and therefore needs to model only a small subset of the biological encoding space, which can be covered by a smaller number of training samples. When we re-trained with 100,000 train images instead of 50,000, the improvement in surrogate validation errors and overall prediction errors was negligible. However, we cannot rule out that more data or advanced training techniques such as adversarial-resistant training would improve the surrogates performance (now acknowledged on lines 294-295).
>
> > *I know this is not the contribution of this work, but a bit more discussion of where the patient-specific data comes from in the text would be helpful for the reader.*
>
> We thank the reviewer for their suggestion. We have extended Section 2 to give a bit more background on how patient-specific data were acquired through psychophysical tasks such as drawings, brightness ratings, and size ratings (lines 79-81), the full details of which are described in [21, 23].

---

### Official Review · Reviewer_UeDa · 2022-07-11

**Rating:** 7
**Confidence:** 4
**Soundness:** 3 good
**Presentation:** 3 good
**Contribution:** 3 good

**Summary:**

The manuscript presents a encoder to generate optimal stimulation parameters for a visual prosthesis. Specifically, the focus of the paper is on developing an end-to-end deep learning model that is trained to invert a known fixed-forward model.

**Questions:**

Please check the weakness and strengths.

**Limitations:**

Check Weakness.

**Strengths And Weaknesses:**

Strength
1. The idea is quite interesting on using HNA for mapping input images to stimulation parameters.
2. It can increase the accuracy of retinal prosthesis and in general other sensory feedback systems.

Weakness
 There are several questions that I would invite the authors to address

1.	The results and metrics are based on a forward model designed in 19. If the forward model changes, would you need to update the encoder?
2.	If the forward model was not fixed how would the HNA perform? From a real retinal implant perspective due to plasticity in visual cortex would the stimulation parameters learnt using a fixed decoder change?
3.	After Figure 1C can there be a Figure 1D to show how would this be deployed since there will not real time perceptual loss in case of patients (maybe some qualitative metrics but a quantitative loss)?
4.	Figure 3 can the authors add how the stimulation parameters for electrodes looks like for a given MNIST image. Say for 0 input what is the output of the gride of an electrode while using HNA, Surrogate and Naïve.
5.	What is computational burden of these Encoder? Since they are for implantable applications this would be an important metric. Latency for such computing these models. Definitely cannot use NVIDIA GPU for the implanted application.  How would this scale.
Minor issues:
Line 100 is not formatted correctly.

---

> ### Author Response · Authors · 2022-08-02
> **Author Response to Reviewer UeDa**
>
> We are grateful to the reviewer for their positive review, and their thoughtful questions and suggestions.
>
> > *1. The results and metrics are based on a forward model designed in 19. If the forward model changes, would you need to update the encoder?*
>
> The short answer is yes: If the forward model were to change, then the encoder would have to be retrained with the new forward model as decoder. However, for epiretinal prostheses, it is unlikely that the chosen phosphene model would require changes, because it can already account for the range of reported percepts through different patient-specific parameters $\phi$.
>
>
>
> > *2. If the forward model was not fixed how would the HNA perform? From a real retinal implant perspective due to plasticity in visual cortex would the stimulation parameters learnt using a fixed decoder change?*
>
> A dynamic forward model either means that the patient-specific parameters $\phi$ for an individual have changed, or the chosen forward model is no longer applicable and must be replaced. For the first case, the model requires no changes and the new $\phi$ can be measured and input to the existing encoder. In the second case, the encoder would have to be retrained with the new decoder. However, most existing literature in visual prostheses suggests that phosphenes remain stable over time, and that there is minimal associated plasticity in the early visual system [24, 25] (lines 79-81).
>
> > *3. After Figure 1C can there be a Figure 1D to show how would this be deployed since there will not real time perceptual loss in case of patients (maybe some qualitative metrics but a quantitative loss)?*
>
> We thank the reviewer for their suggestion and have added a panel for figure 1D showing HNA deployment for visual prostheses.
>
> > *4. Figure 3 can the authors add how the stimulation parameters for electrodes looks like for a given MNIST image. Say for 0 input what is the output of the gride of an electrode while using HNA, Surrogate and Naïve.*
>
> This is a great suggestion, which we have addressed in Appendix D. There we present an analysis of the predicted stimuli that may answer any related questions. Due to space limitations, we were unable to include this analysis in the main text of the original manuscript. The appendices are included in the rebuttal revision PDF and the supplemental material of the original submission.
>
> > *5. What is computational burden of these Encoder? Since they are for implantable applications this would be an important metric. Latency for such computing these models. Definitely cannot use NVIDIA GPU for the implanted application.  How would this scale.*
>
> We agree that further discussion on deployment is necessary, and have added a deployment section in the Discussion to address this (lines 299-303). To summarize, the encoder inference time for a single target MNIST image is 1.2 ms on a Nvidia 3090 GPU and 4.1ms on a single-core 2.1GHz CPU, and the saved model takes 5.3MB of memory. Future work could reduce these measures by employing techniques such as pruning, mixed floating point precision, search over smaller encoder architectures, or Tensorflow Lite conversion. Another practical solution may come in the form of Edge AI accelerators (e.g., Intel’s Neural Compute Stick) and deep learning-compatible neuromorphic hardware (e.g., BrainChip’s Akida Neuromorphic SoC), which can lead to orders of magnitude reductions in size, weight, and power compared to conventional CPUs or GPUs.

---

### Official Review · Reviewer_cH4d · 2022-07-11

**Rating:** 7
**Confidence:** 4
**Soundness:** 3 good
**Presentation:** 3 good
**Contribution:** 3 good

**Summary:**

This paper develops an autoencoder to solve the inverse problem of converting incoming visual stimuli into electrical stimulation patterns such that the responses evoked by the electrical stimulation mimics the target visual stimulus.

By using an accurate decoding model, perceptually accurate loss function and training across multiple participants, the encoder outperforms existing approach for second-sight retinal prosthesis.

**Questions:**

1. Is it hard to directly solve Eq. 1? Since the number of parameters in $\mathbf{s}$ is not large (15 x 15 x 3), it might be possible to solve Eq. 1 directly. Why do you need to learn an amortized encoder? Atleast this will give an upper bound on HNA performance.

2. How accurately are the patient specific parameters ($\phi$ ) known? What are the consequences of inaccuracies in their estimation? Like the failure of surrogate model, does HNA fail if encoder and decoder have different estimates of $\phi$ ? If  $\phi$ is estimated incorrectly / changes over time, then this question becomes relevant.



**Limitations:**

Limitations are described adequately.

**Strengths And Weaknesses:**

Strengths:

1. Well written.
2. Advances state-of-the-art in the artificial retina application and shows how their approach is better using overall convincing analysis.

Weakness:
1. A lot of innovation (particular architecture of encoder/decoder) relies on specifics of crude retinal prosthesis. I am not sure how general purpose the approach presented here is. Does the model of phosphenes translate to other neural stimulation systems like somatosensory / spinal cord stimulation?

2. The approach only works for static images as visual targets. How does the autoencoder framework extend to dynamics targets?

3. The decoder description on page 5 is short. Might help to improve it for broader applicability.

---

> ### Author Response · Authors · 2022-08-02
> **Author Response to Reviewer cH4d**
>
> We thank the reviewer for their encouraging evaluation and helpful feedback.
>
> **Weaknesses**
> > 1. A lot of innovation (particular architecture of encoder/decoder) relies on specifics of crude retinal prosthesis. I am not sure how general purpose the approach presented here is. Does the model of phosphenes translate to other neural stimulation systems like somatosensory / spinal cord stimulation?
>
> We agree that both the phosphene model and the learned stimulus encoder presented here are specific to retinal prostheses. The underlying HNA technique, however, is more general, and it seems likely that other neuroprostheses may benefit from HNA’s end-to-end treatment of stimulus encoding. We envision it providing a framework for researchers developing other sensory neuroprostheses, motivating development of differentiable forward models of biological systems for use in HNA-like models. The performance of this approach will likely vary across devices, but HNA should apply wherever there is open-loop control and a clear target response (i.e., no feedback is used to constrain subsequent stimuli, and there exists a goal response state that can be represented and modeled), such as for auditory, somatosensory, and possibly spinal cord implants. We have updated our Broader Impacts subsection to address these points (lines 304-311).
>
> > 2. The approach only works for static images as visual targets. How does the autoencoder framework extend to dynamics targets?
>
> The reviewer is correct that HNA as presented here is only applicable to static targets, and therefore dynamic targets must be split into individual frames and encoded separately. High-fidelity stimulus encoding for dynamic targets is an open challenge for visual prostheses, and likely requires further understanding of the temporal dynamics of both the electrode-retina interface and the early visual system. However, one straightforward adaptation might be to encode entire stimulus sequences as opposed to single stimuli that are optimized to reconstruct the dynamic target sequence (lines 312-315).
>
> > 3. The decoder description on page 5 is short. Might help to improve it for broader applicability.
>
> We agree, and have updated the decoder description to clarify that this specific decoder is only for retinal prostheses, and to give examples of what decoders might look like  for other neuroprostheses. (lines 151-155).
>
> **Questions**
>
> > 1. Is it hard to directly solve Eq. 1? Since the number of parameters in **s** is not large (15 x 15 x 3), it might be possible to solve Eq. 1 directly. Why do you need to learn an amortized encoder? Atleast this will give an upper bound on HNA performance.
>
> To the best of our knowledge, the nonlinearity and complexity of both the phosphene model decoder and the perceptual objective function prohibit a closed-form solution to Eq. 1. Instead, SGD could be used to find the optimal solution for each individual target image. However, in our preliminary experiments this took prohibitively long and would thus be infeasible to be deployed in a real system. An amortized encoder has the advantage of providing a fast, approximately optimal stimulus. We do agree that this could provide an insightful upper bound on HNA performance, but unfortunately would have required more compute resources and time than available.
>
> > 2. How accurately are the patient specific parameters ($\phi$) known? What are the consequences of inaccuracies in their estimation? Like the failure of surrogate model, does HNA fail if encoder and decoder have different estimates of  $\phi$? If $\phi$ is estimated incorrectly / changes over time, then this question becomes relevant.
>
> The reviewer raises an excellent point. In general, there may be some uncertainty in $\phi$. We thus include an additional analysis on the consequences of mis-specified $\phi$ on the produced encodings in Appendix H. In short, we found that even with large errors in $\phi$, the HNA encoder still outperforms the naive encoder in nearly all cases. Additionally, for retinal prostheses, $\phi$ can be estimated with reasonable accuracy using the methods described in [21, 23], and generally does not change dramatically over time [24, 25] (lines 79-81).

---

### Author Response · Authors · 2022-08-02
**General Response to Reviews**

We are very grateful to the reviewers for their thoughtful feedback and questions, which have been extremely valuable in understanding how to improve our paper. We are pleased that reviewers agreed HNA was an effective solution and improved state of the art in visual prostheses (R1 cH4d , R2 UeDa , R3 kHVG). We are glad they thought our treatment of the problem to be very elegant (R3), found the analysis convincing and paper well written (R1), and were optimistic the results might generalize to other sensory feedback systems (R2). Reviewers raised insightful questions about the model’s adaptability to changes in the forward model (R2), limitations of the baseline surrogate model (R3), and direct solution of the objective function (R1). We also found their suggestions to be very helpful, especially regarding analysis and discussion of patient-specific parameters (R1, R3), discussion of deployment capabilities on resource constrained devices (R2), and additional figures (R2).

Below we address each of the reviewers’ questions and concerns point by point. We have also uploaded a new rebuttal revision PDF, which integrates their feedback and includes the appendices (previously only in supplemental material), updated with one additional section addressing reviewers’ questions on patient-specific parameters (Appendix H). To aid reviewers, we have marked our new revisions in blue.

---

### Meta-Review · Area_Chair_oMWD · 2022-08-23

**Recommendation:** Accept
**Confidence:** Certain

**Metareview:**

This paper formulates the problem of learning how to stimulate a visual neuroprosthesis as a hybrid autoencoder.  While the decoder can be taken as a known and fixed model that describes how stimuli produce percepts, the encoder needs to be learned.  Once learned the encoder maps target percepts into stimuli that can be passed into the device (decoder).

Motivation and formulation of the problem is especially clear / strong.  The paper is well written and the reviewers and I appreciated the nice solution strategy for a potentially impactful application area.  There were some concerns about how generally applicable the approach is.  However, the results presented likely do advance the state of the art in this setting.

Given my own reading of the paper and the consistently positive reviewer scores, I'm very comfortable endorsing this paper for acceptance.

**Award:**

No

---

### Decision · Program_Chairs · 2022-09-14

Accept